# Clinical Implication of Drug Resistance for *H. pylori* Management

**DOI:** 10.3390/antibiotics11121684

**Published:** 2022-11-23

**Authors:** Erick A. Argueta, Jonathan J. C. Ho, Yousef Elfanagely, Erika D’Agata, Steven F. Moss

**Affiliations:** 1Division of Gastroenterology, Rhode Island Hospital, Warren Alpert Medical School of Brown University, Providence, RI 02903, USA; 2Department of Medicine, Rhode Island Hospital, Warren Alpert Medical School of Brown University, Providence, RI 02903, USA; 3Division of Infectious Diseases, Rhode Island Hospital, Warren Alpert Medical School of Brown University, Providence, RI 02903, USA

**Keywords:** *Helicobacter pylori* 1, resistance 2, antimicrobial susceptibility testing 3

## Abstract

Rates of antimicrobial-resistance among *H. pylori* strains are increasing worldwide, resulting in declining eradication rates with current therapies, especially those containing clarithromycin or levofloxacin. To improve *H. pylori* management, a paradigm shift is needed, from the empiric approaches formerly employed, to regimen selection based upon knowledge of local and patient-level antimicrobial susceptibility data. We review the mechanisms of *H. pylori* antimicrobial resistance and the available worldwide pattern of resistance to key antimicrobials used in *H. pylori* therapy. The practicalities and challenges of measuring susceptibility in clinical practice is discussed, including not only conventional culture-based techniques but also novel sequencing-based methods performed on gastric tissue and stool samples. Though clinical trials of “tailored” (susceptibility-based) treatments have yet to show the clear superiority of tailored over empiric regimen selection, the ability to measure and modify treatment based upon antimicrobial susceptibility testing is likely to become more frequent in clinical practice and should lead to improved *H. pylori* management in the near future.

## 1. Introduction

*Helicobacter pylori* (*H. pylori*) infection has a well-established role in peptic ulcer disease, gastric cancer, and mucosa-associated lymphoid tissue lymphoma. *H. pylori* eradication cures gastritis and can alter the progression of long-term complications or the recurrence of disease [1]. Eradication therapies have evolved since *H. pylori* was first cultured 40 years ago. Early on, it was appreciated that single antimicrobials had very low rates of eradication but that combination therapies were more successful. In the 1990s, proton pump inhibitors (PPIs) were also included in eradication regimens because acid suppression was found to increase eradication rates, in addition to benefiting ulcer healing [2]. 

Currently available therapies reflect the results of many clinical trials that have been performed without regard to susceptibility testing. The success rates of these therapies is highly variable around the world, reflecting differing population adherence, pharmacokinetics, and antimicrobial susceptibility (Table 1). Rates of antimicrobial-resistance among *H. pylori* strains are increasing worldwide. In this paper, mechanisms and rates of resistance and implications in the treatment of *H. pylori* infections will be reviewed. The availability of rapid and accurate sequencing-based methods to determine antimicrobial-resistance profiles among *H. pylori* strains and its benefits in choosing the optimal antimicrobials to include in treatment regimens is also discussed. 

## 2. Worldwide Rates of *H. pylori* Antimicrobial Resistance 

There have been several recent reviews and meta-analyses assessing *H. pylori* resistance rates to the antimicrobials that are commonly used in eradication regimens [3,4,5]. Although the availability and quality of such data is highly variable and is scant or non-existent for some regions, the general trend has been a steady rise in reported resistance to clarithromycin, levofloxacin and metronidazole over the last four decades since *H. pylori* resistance was first recognized. Figure 1 illustrates the most recent reported resistance rates by region (Figure 1A–E). 

For clarithromycin, resistance rates have been reported to be as low as 0% in Gambia, Ethiopia, Canada, and Bhutan to as high as 96% in Australia. Resistance to metronidazole has been reported to be as low as 1% in Iceland to 100% in India. Globally, resistance rates for rifabutin and levofloxacin vary from 0% to 96.8% and 0% to 66%, respectively. From 2006–2016, data show progressively increasing rates of resistance to both clarithromycin and metronidazole from 15% to 23% in Europe and from 4% to 14% in the western Pacific region [4]. Table 2, Table 3, Table 4, Table 5, Table 6 and Table 7 list country, territory, or region with its associated rate of resistance by antibiotic. Increased resistance is related to greater community consumption of the specific antimicrobial for all indications, not just for *H. pylori* eradication. For example, across Europe, there is a significant association between general macrolides and fluoroquinolone consumption and *H. pylori* resistance to clarithromycin and levofloxacin; a study from Israel found an association between increased cumulative exposure to macrolides and a decreasing likelihood of eradication success for clarithromycin triple therapy [6,7].

A recent meta-analysis from the United States reported resistance rates of over 30% for clarithromycin, levofloxacin, and metronidazole over the past decade and a dual clarithromycin–metronidazole resistance rate of 11.7% [3]. However, it was recognized that these data are based on a small sample of total US cases, with a skewed geographic distribution of medical centers, representing Alaska, California, Minnesota, New York, Rhode Island, Texas, and Washington. Furthermore, since data were not collected systematically in the US, the role of prior antimicrobial exposure and resistance is unclear.

In contrast to these high and increasing resistance rates for clarithromycin, levofloxacin, and metronidazole for many parts of the world, amoxicillin, tetracycline, and rifabutin resistance has remained low (<5%) in many regions, including Europe and North America. Consequently, these three antimicrobials are increasingly being recommended in regimens in countries where resistance rates are high (>15%) and eradication rates suboptimal. We note that high rates of resistance to amoxicillin and tetracycline have been reported from some countries in Africa, but these reports are inconsistent across the continent for unclear reasons and may be due to methodological issues [5].

## 3. Acquisition of Antimicrobial Resistance among *H. pylori* Strains 

The two main methods of developing antimicrobial-resistant pathogens are endogenous and exogenous acquisition [8]. Endogenous acquisition mainly occurs within an individual via the emergence of resistance in a previously susceptible strain due to antimicrobial exposure and the horizontal transfer of resistance genes [9]. These mechanisms can lead to single and multidrug resistance. Heteroresistance, the presence of *H. pylori* strains that exhibit increased levels of resistance to specific antimicrobials within the gastric population, has also been described [10]. Exogenous acquisition occurs via cross-transmission between persons or a common environmental exposure source. Small older studies have shown the occurrence of clustering in families of clonally related *H. pylori* strains and acquisition outside of the family, implying the exogenous transmission of *H. pylori* strains [11,12]. Endogenous acquisition likely plays the major role in the acquisition of antimicrobial-resistant *H. pylori* strains. A greater understanding of the contribution of these mechanisms of acquisition would have important implications in developing and implementing strategies aimed at decreasing infections caused by antimicrobial-resistant *H. pylori* strains.

## 4. Molecular Mechanisms of Drug Resistance 

Endogenous *H. pylori* resistance primarily arises from genetic changes, which modify drug targets or disrupt the activation of the drug within the cell, resulting in antimicrobial resistance [10]. In resistant strains of *H. pylori*, the genetic changes observed usually occur chromosomally, as opposed to extrachromosomally (e.g., plasmids). Additionally, the genetic changes that typically result in antimicrobial resistance are from genetic mutations rather than resistant gene acquisition. Examples of genetic mutations include missense, nonsense, insertion, or deletions [10]. Less established are the mechanisms of resistance associated with efflux pumps and biofilm formation [13]. Efflux pump systems in bacteria can eject medication and have a critical role in resistance. Studies have described specific efflux pump genes implicated in *H. pylori* resistance, though the mechanism of acquisition is unclear. Similarly, biofilm formation has an unclear mechanism of acquisition but confer greater resistance to medication by increasing *H. pylori*’s ability to survive a harsher external environment [14].

### 4.1. Clarithromycin Resistance 

Macrolides such as clarithromycin inhibit bacterial growth by interfering with protein synthesis through binding to the 23S ribosomal subunit. The vast majority of clarithromycin resistance in *H. pylori* has been attributed to an A to G transitional point mutation at position 2143 (A2143G) or an A to G or C mutation at position (A2142G/C) in the 23S rRNA gene preventing clarithromycin from inhibiting protein synthesis [10,15].

### 4.2. Levofloxacin Resistance 

Levofloxacin is a fluoroquinolone drug that directly inhibits bacterial DNA synthesis via the inhibition of DNA gyrase, which is essential for the replication and transcription of DNA [16]. Mutations in the genes that encode DNA gyrase (*gyrA* and *gyrB* genes) account for most of the levofloxacin-resistant *H. pylori* strains [17,18]. Miyachi et al. noted that point mutations at *gyrA* at Asn-87 and Asp-91 had a critical impact on the levofloxacin resistance of *H. pylori* in Japan [18]. Several other point mutations in the *gyrA* gene have been identified, which confer levofloxacin resistance in *H. pylori* strains, with *gyrB* mutations described less commonly. The few specific mutations that are associated with most cases of phenotypic resistance means that molecular resistance testing for levofloxacin, as with clarithromycin, is a reliable substitute for culture and sensitivity testing.

### 4.3. Metronidazole Resistance 

Metronidazole enters the cell as a prodrug via passive diffusion [19]. In the cytoplasm of bacteria, metronidazole is converted into its cytotoxic form by intracellular reduction. The activated reduced metronidazole molecule binds nonspecifically to DNA, which leads to DNA damage and cell death [19]. Multiple different mechanisms leading to metronidazole resistance from reduced nitroreductase activity have been described for *H. pylori* strains, most of which are in the *rdxA* gene and, to a lesser extent, the *frxA* gene [20]. Low levels of reductase enzymes compromise the activation of metronidazole, which is necessary for cytotoxicity. Additional mechanisms have been implicated in metronidazole resistance, including mutations in other enzymes involved in DNA repair and in metronidazole drug efflux. The complexity of metronidazole metabolism and the diversity of identified mutations likely explain the relatively poor correlation of the results of metronidazole resistance testing with its efficacy in *H. pylori* eradication regimes clinically. Unlike the profound negative effects of clarithromycin or levofloxacin resistance on the outcomes of triple therapy with regimens that contain these drugs, metronidazole-resistant *H. pylori* can still be successfully eradicated with metronidazole containing bismuth quadruple therapy, especially with higher metronidazole dosing, albeit with a reduction in the eradication rate from approximately 80 to 70% [21]. 

### 4.4. Amoxicillin Resistance

Bacterial resistance to amoxicillin and other penicillins usually occurs through upregulating beta-lactamase expression. In contrast, resistance to amoxicillin in *H. pylori* strains is mainly due to one of several mutations in the penicillin binding motif of Penicillin-Binding Protein 1A, which decreases the affinity of amoxicillin to its binding site to inhibit cell wall synthesis [22]. Mutations in other penicillin-binding proteins have also been described, as have mutations in membrane proteins that alter membrane permeability to amoxicillin [10].

### 4.5. Tetracycline Resistance

Similar to the mechanism of clarithromycin resistance, the mechanism of tetracycline resistance can be a consequence of mutations that affect tetracycline’s ability to bind to the ribosome. Tetracycline binds to the 30S subunit of the ribosome to stall the synthesis of peptides [23]; mutations in the 16S rRNA affect the binding site of tetracycline. Triple-base pair 16S rDNA mutations, such as AGA to TTC at positions 926–928, confer high tetracycline resistance in *H. pylori* [17]. Other mechanisms of tetracycline resistance in *H. pylori* include increased drug efflux and decreased membrane permeability, as described in some amoxicillin-resistant *H. pylori* strains [24].

### 4.6. Rifabutin Resistance

Rifabutin and other drugs of the rifamycin class inhibit bacterial transcription through binding to the beta subunit of bacterial DNA-dependent RNA polymerase, encoded by the *rpoB* gene. Rifabutin resistance is rare and usually due to one of several specific mutations in *rpoB* [25].

## 5. Methods to Detect Resistance

The decline in eradication rates over the last 20 years rates throughout the world, as a result of increasing antimicrobial resistance, has led to increased awareness of, and demand for, resistance testing, especially for refractory cases [26]. Two major types of testing are available—traditional microbiological techniques that require transport to a laboratory equipped for culturing *H. pylori* and direct antimicrobial resistance testing, using molecular-based techniques such as polymerase chain reaction (PCR)-based technologies and next-generation sequencing (NGS). The pros and cons of the various methodologies and the practicalities of using resistance testing in the US have recently been reviewed [27].

Historically, *H. pylori* could best be cultured using fresh gastric biopsies rapidly plated onto suitable media in a local dedicated (often research) laboratory. However, samples can also be inoculated in transport media with samples immediately frozen prior to shipment to an outside laboratory [27]. Successful isolation and cultivation of *H. pylori* using this approach has remained challenging, costly, and affected by several factors, including the time interval between specimen collection and inoculation, as well as inappropriate transport conditions. Consequently, culture-based resistance has generally been underutilized in clinical practice, although it is available in the United States from several commercial laboratories.

Molecular susceptibility testing overcomes the practical challenges of culturing *H. pylori* and instead probes for the resistance-associated mutations that underlie phenotypic resistance. PCR detection is available for clarithromycin and levofloxacin in gastric tissues and in stool in some countries. The ability to detect resistance from stool samples is especially valuable for patients in whom endoscopy is not otherwise clinically indicated. A more comprehensive approach to resistance profiling can be obtained by NGS, which permits the simultaneous evaluation of multiple-resistance-associated mutations to the six most commonly prescribed antimicrobials in *H. pylori* treatment regimens: clarithromycin, amoxicillin, tetracycline, metronidazole, rifabutin, and levofloxacin [28]. NGS is also particularly applicable to the detection of heteroresistance within *H. pylori* communities, as multiple sequencing reads are obtained per sample. NGS can be performed on fresh, frozen, or archived formalin-fixed paraffin-embedded gastric tissue blocks [29,30], and recent advances in specimen processing can now reliably detect and sequence *H. pylori* DNA and reflectively test for resistance in stool samples, bypassing the inconvenience, costs, and risks of endoscopy [27,31].

While NGS can quickly and accurately provide sequence information, translating its output to susceptibility profiling depends upon the level of understanding of how closely antimicrobial gene mutations correlate with phenotypic resistance. NGS may underestimate resistance from unappreciated (novel) mutations or from resistance caused via nongenetic mechanisms. In addition, NGS cannot determine minimum inhibitory concentrations (MIC) and cannot distinguish between active infection or simply the presence of *H. pylori* DNA colonization. On the other hand, NGS is very sensitive to mutation detection, whereas traditional culture techniques sometimes fail to grow viable *H. pylori* colonies from *H. pylori*-positive patients, especially if they were exposed to proton pump inhibitors. Cultures are successful in less than 80% of cases after one unsuccessful attempt at eradication [32]. Clarithromycin and levofloxacin resistance determined by NGS appear to be highly predictive of eradication failure with triple therapy. In contrast, this correlation may be less robust for amoxicillin and metronidazole [29,30]. Given the rarity of tetracycline and rifabutin resistance, the clinical significance of NGS testing for these antimicrobials remains to be formally established. Further prospective evaluation of phenotypic resistance predicted by NGS, resistance evaluated by traditional methods and ultimately the eradication success of regimens chosen based on these techniques is needed, as are cost-effectiveness studies.

## 6. The Role of Antimicrobial Susceptibility Testing in Clinical Practice

The treatment of *H. pylori* has for many years been empiric and therefore an outlier in infectious disease management, wherein antimicrobials are tailored to the antimicrobial-susceptibility pattern of the causative pathogen. Declining eradication rates due to antimicrobial resistance have now prompted a reevaluation of this empiric approach and an increasing enthusiasm for the adoption of treatments based on the principles of infectious disease management and antimicrobial stewardship. The standard practice of treating infectious diseases for the great majority of scenarios dictates antimicrobial susceptibility testing for the causative pathogen. Once results are available, the antimicrobials are tailored to optimize treatment and reduce unnecessary antimicrobial exposure [33]. Since treating *H. pylori* is never an emergency, selecting a regimen based on knowledge of local susceptibility patterns or, ideally, on the susceptibility profile of an individual’s infecting strain should be a more optimal approach, as it would prevent patient exposure to antimicrobials that are ineffective against a resistant *H. pylori* strain.

### 6.1. Before Primary Treatment (Treatment-Naïve Patients)

Most guidelines recommend avoiding clarithromycin and levofloxacin in empiric regimens in regions where the local resistance rate is above a somewhat arbitrary 15% [34,35]. Since this is the case for most of the world, including the United States (albeit based on limited data) [3], this limits empiric first-line regimens to some form of bismuth quadruple therapy (bismuth, acid suppression medication, plus two of metronidazole, tetracycline, and/or amoxicillin) or, in certain countries, a rifabutin–amoxicillin–proton-pump-inhibitor (PPI) combination or a PPI or vonoprazan–amoxicillin dual therapy.

**Table 1 antibiotics-11-01684-t001:** Some common therapies used for *H. pylori* eradication.

Therapy	Typical Regimen	Comments
Bismuth quadruple	PPI (standard dose) BID, bismuth subcitrate (120–300 mg) or subsalicylate (300 mg) QID, tetracycline (500 mg) QID, metronidazole (250–500 mg) QID for 10 to 14 days	Currently the most efficacious empiric therapy [34]. However, bismuth is not available in all countries. Adherence is challenging, improved by multi-drug single-capsule formulations.
Clarithromycin triple	PPI (standard or double dose) BID, clarithromycin (500 mg), amoxicillin (1 g), or metronidazole (500 mg TID) for 10 to 14 days	Still the most prescribed in the US, but efficacy has been falling continuously, due to increasing clarithromycin resistance.
Levofloxacin triple	PPI (standard dose) BID, levofloxacin (500 mg) QD, amox (1 g) BID for 10 to 14 days	Formerly a useful second-line regimen, but rising levofloxacin resistance rates has led to decreased efficacy.
Rifabutin triple	PPI (standard dose) BID, rifabutin (300 mg) QD, amox (1 gm) BID for 10 days	Useful for first-line therapy or in refractory cases, since rifabutin and amoxicillin resistance remain rare [29].
Vonoprazan-containing regimens	Vonoprazan 20 mg, amoxicillin 1000 mg, clarithromycin 500 mg, each given twice daily for 14 days	Substituting vonoprazan for PPIs produces greater acid suppression and increased eradication rates.

**Table 2 antibiotics-11-01684-t002:** North America Regional Rates of Resistance.

Country/Territory/Region Name	Clarithromycin	Amoxicillin	Metronidazole	Tetracycline	Rifabutin	Levofloxacin
Alaska	33.8%	3.1%	46.2%	0.0%	N/A	23.6%
California	26.7%	N/A	N/A	N/A	N/A	N/A
Minnesota	70.4%	1.2%	82.4%	1.7%	N/A	N/A
New York	29.3%	6.7%	27.5%	0.0%	N/A	52.0%
Rhode Island	30.2%	1.1%	33.3%	0.5%	0.5%	29.6%
Texas	14.6%	0.0%	28.3%	0.6%	N/A	29.1%
Washington	44.1%	N/A	N/A	N/A	N/A	N/A
Canada	0.0%	N/A	12.0%	N/A	N/A	N/A

N/A: data not available. Adapted from [3,4,5].

**Table 3 antibiotics-11-01684-t003:** South America Regional Rates of Resistance.

Country/Territory/Region Name	Clarithromycin	Amoxicillin	Metronidazole	Tetracycline	Rifabutin	Levofloxacin
Argentina	7.0%	2.0%	8.0%	N/A	N/A	9.0%
Brazil	16.0%	7.0%	40.0%	N/A	N/A	10.0%
Colombia	6.0%	4.0%	N/A	N/A	N/A	N/A
Peru	36.0%	33.0%	62.0%	4.0%	N/A	36.0%

N/A: data not available. Adapted from [3,4,5].

**Table 4 antibiotics-11-01684-t004:** Europe Regional Rates of Resistance.

Country/Territory/Region Name	Clarithromycin	Amoxicillin	Metronidazole	Tetracycline	Rifabutin	Levofloxacin
Austria	32.0%	0.0%	14.0%	0.0%	N/A	11.0%
Belgium	36.0%	N/A	40.0%	N/A	N/A	29.0%
Bulgaria	23.0%	1.0%	28.0%	3.0%	N/A	10.0%
Croatia	14.0%	0.0%	11.0%	N/A	N/A	2.0%
France	43.0%	0.0%	65.0%	0.0%	N/A	15.0%
Germany	13.0%	0.0%	37.0%	0.0%	N/A	18.0%
Greece	36.0%	0.0%	38.0%	0.0%	N/A	10.0%
Ireland	31.0%	0.0%	31.0%	0.0%	N/A	11.0%
Italy	15.0%	0.0%	26.0%	0.0%	N/A	5.0%
Spain	19.0%	0.0%	N/A	0.0%	N/A	16.0%
Netherlands	16.0%	N/A	15.0%	N/A	N/A	N/A
United Kingdom	36.0%	2.0%	57.0%	2.0%	N/A	11.0%
Poland	N/A	0.0%	38.0%	0.0%	N/A	8.0%
Iceland	6.0%	0.0%	1.0%	0.0%	N/A	6.0%

N/A: data not available. Adapted from [3,4,5].

**Table 5 antibiotics-11-01684-t005:** Middle East Regional Rates of Resistance.

Country/Territory/Region Name	Clarithromycin	Amoxicillin	Metronidazole	Tetracycline	Rifabutin	Levofloxacin
Iran	21.0%	14.0%	63.0%	15.0%	N/A	25.0%
Pakistan	40.0%	25.0%	62.0%	12.0%	N/A	24.0%
Saudi Arabia	23.0%	15.0%	49.0%	N/A	N/A	13.0%
Israel	47.0%	4.0%	57.0%	2.0%	N/A	5.0%
Turkey	28.0%	N/A	35.0%	N/A	N/A	30.0%

N/A: data not available. Adapted from [3,4,5].

**Table 6 antibiotics-11-01684-t006:** Africa Regional Rates of Resistance.

Country/Territory/Region Name	Clarithromycin	Amoxicillin	Metronidazole	Tetracycline	Rifabutin	Levofloxacin
Algeria	26.0%	0.0%	56.3%	0.5%	N/A	0.0%
Morocco	27.4%	0.0%	40.0%	0.0%	N/A	0.0%
Senegal	2.2%	0.0%	81.8%	0.0%	N/A	15.0%
Gambia	0.0%	N/A	29.0%	N/A	N/A	N/A
Burkina Faso	9.1%	N/A	N/A	N/A	N/A	N/A
Cameroon	29.2%	91.4%	95.6%	2.9%	N/A	0.0%
Nigeria	36.0%	67.7%	94.2%	58.0%	96.8%	0.0%
South Africa	18.4%	21.5%	91.3%	8.7%	N/A	10.3%
Congo	12.8%	34.3%	90.2%	2.5%	N/A	57.5%
Tanzania	28.7%	N/A	58.8%	N/A	N/A	N/A
Kenya	6.4%	4.6%	52.3%	N/A	N/A	N/A
Rwanda	13.0%	N/A	36.0%	N/A	N/A	N/A
Sudan	41.1%	55.8%	19.2%	36.5%	N/A	N/A
Mauritania	5.3%	N/A	N/A	N/A	N/A	N/A
Ethiopia	0.0%	3.0%	85.4%	0.0%	0.0%	N/A
Libya	N/A	N/A	70.3%	N/A	N/A	N/A
Tunisia	15.4%	0.0%	51.3%	N/A	N/A	N/A
Egypt	56.0%	N/A	63.0%	N/A	N/A	N/A

N/A: data not available. Adapted from [3,4,5].

**Table 7 antibiotics-11-01684-t007:** Asia and Oceania Regional Rates of Resistance.

Country/Territory/Region Name	Clarithromycin	Amoxicillin	Metronidazole	Tetracycline	Rifabutin	Levofloxacin
Bangladesh	39.0%	4.0%	95.0%	0.0%	N/A	66.0%
Bhutan	0.0%	0.0%	83.0%	0.0%	N/A	5.0%
India	5.0%	65.0%	100.0%	0.0%	N/A	N/A
Indonesia	9.0%	5.0%	47.0%	3.0%	N/A	31.0%
Thailand	19.0%	1.0%	44.0%	0.0%	N/A	19.0%
Australia	96.0%	0.0%	68.0%	0.0%	N/A	5.0%
China	37.0%	1.0%	77.0%	2.0%	N/A	33.0%
Japan	28.0%	N/A	N/A	N/A	N/A	N/A
Laos	13.0%	N/A	N/A	N/A	N/A	13.0%
Malaysia	5.0%	0.0%	82.0%	12.0%	N/A	N/A
New Zealand	7.0%	5.0%	49.0%	0.0%	N/A	N/A
Singapore	16.0%	4.0%	44.0%	7.0%	N/A	13.0%
South Korea	18.0%	4.0%	40.0%	4.0%	N/A	28.0%
Taiwan	26.0%	1.0%	31.0%	2.0%	N/A	15.0%
Vietnam	63.0%	2.0%	58.0%	17.0%	N/A	32.0%

N/A: data not available. Adapted from [3,4,5].

Two meta-analyses have shown a benefit in increasing eradication rates when susceptibility-guided treatment was performed [36,37]. However, both demonstrated that better efficacy results occurred only among patients in whom the susceptibility-guided strategy, as opposed to empiric therapy, was implemented for first-line triple regimens—combinations that are no longer considered an ideal empiric regimen. In contrast, a difference was not demonstrated for quadruple-based regimens, which may reflect the increased likelihood of susceptibility among the greater number of drugs used in the regimen.

The role of resistance testing prior to rifabutin-based primary therapies has not been evaluated but is unlikely to be helpful currently because of low baseline amoxicillin and rifabutin resistance in most populations. Similar considerations apply to vonoprazan or PPI-based dual therapies.

### 6.2. After Treatment Failure (Refractory Patients)

Clinical practice guidelines recommend susceptibility testing once a patient has failed one or more empiric treatment options, particularly if they have a true penicillin allergy [26,34]. Selecting a regimen based on the results of susceptibility testing is logical and theoretically should be superior to a “best guess” based on prior antimicrobial history, including antimicrobials taken for other types of infections and knowledge of local resistance profiles.

Several studies have been performed to compare tailored versus empiric treatments for refractory infection. For example, in a multi-center trial from Taiwan on *H. pylori* refractory to at least two previous treatments, culture-guided susceptibility testing on gastric biopsies was not significantly superior to empirically chosen sequential regimens based on medication history (78% vs. 72.2%, respectively) [38].

A recent comprehensive meta-analysis reviewed 16 studies (with a total of 1283 patients) of tailored versus empiric therapy for refractory infection [36]. Most studies were of second-line therapy, and nine of them were randomized. Overall, there was no significant advantage of the susceptibility-based approach, irrespective of how the data were analyzed (such as randomized versus non-randomized studies, second-line versus third-line treatments). The authors concluded that “The benefit of susceptibility-guided treatment over empirical treatment of *H. pylori* infection could not be demonstrated, either in first-line (if the most updated quadruple regimens are prescribed) or in rescue therapies.”

The results of these randomized trials performed to date are sobering and perhaps counter-intuitive, as they demonstrate a contrast between expert opinion recommending resistance testing for refractory cases [26,34] and data from the literature that it does not improve outcomes. More research is needed in this area since many of the relevant studies have been limited in numbers of patients and rigorousness of study design and largely reflect practice when clarithromycin-based therapies were the empiric first choice.

## 7. Conclusions & Future Perspectives

Our current *H. pylori* eradication regimens are the result of empiric development by gastroenterologists over the last 40 years. Declining *H. pylori* eradication rates worldwide linked to increasing *H. pylori* resistance to clarithromycin, levofloxacin, and metronidazole are now focusing attention on the need to apply a more scientific approach to *H. pylori* treatment, utilizing the principles of infectious disease, epidemiology, and antimicrobial stewardship.

To this end, the ability to measure resistance to the antimicrobials used in current and emerging regimens is critical. On an individual level, it can guide therapy to prevent the complications of chronic *H. pylori* infection, while knowledge of local and regional antimicrobial resistance is essential for selecting and designing rational therapies for populations. The increasing availability of molecular methods, including NGS, and their applicability in stool testing should help overcome the technical difficulties related to conventional culture and susceptibility testing, as well as a dependence on endoscopy for sample collection. However, issues of the cost and availability of such testing are substantial, especially for certain regions of the world where *H. pylori* prevalence is highest.

Importantly, it remains incumbent on the scientific community to prove formally that knowledge of resistance can improve outcomes of *H. pylori* treatments. As described above, most clinical trials have not demonstrated that susceptibility-based (or “tailored”) therapy is superior to currently recommended empiric regimens, either for treatment-naïve or previously treated patients. However, most of these trials have employed suboptimal study designs based on now-redundant regimens, and with relatively small patient numbers. As a result, scientifically robust prospective studies are needed before tailored therapy can be fully embraced.

In the short-term, much can be done to improve *H. pylori* management, including improved systems of collecting data on eradication rates and resistance trends. For example, in the US, most clinicians lack knowledge of the success of the regimens that they use since “tests of cure” are often forgotten or avoided [39]; clarithromycin-based regimens remain still the most frequently used despite high clarithromycin resistance rates [40]; data on resistance is extremely sparse—in a country of about 100 million infected persons, resistance data are only available from fewer than 3000 strains collected over the past decade from only 7 states [3]. The worldwide map of *H. pylori* resistance rates is mostly empty—signifying enormous knowledge gaps (Figure 1).

An important step forward to address these necessities is mounting efforts to collect and disseminate data on resistance and eradication rates through the establishment of large registries. The European registry on the management of *H. pylori* infection exemplifies such an undertaking, which has collated data from over 40,000 patients from 29 countries to date [41]. Similar registries are in development in other global regions and the knowledge obtained from them is likely to help overcome our dependence on the archaic and unscientific empiric *H. pylori* management practices that have been employed in the past.

## Figures and Tables

**Figure 1 antibiotics-11-01684-f001:**
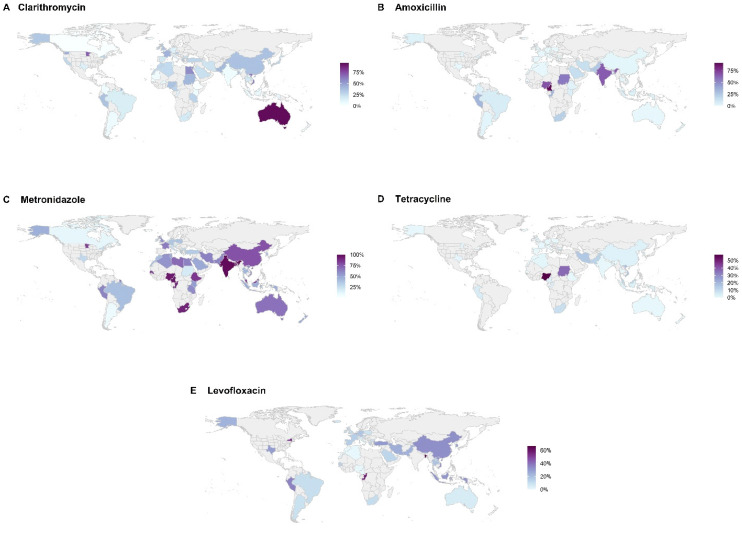
Regional rates of resistance are separated by clarithromycin (**A**), amoxicillin (**B**), metronidazole (**C**), tetracycline (**D**), and levofloxacin (**E**). However, there were insufficient data to adequately illustrate rifabutin resistance. Areas in gray indicate no data available from the references that were used [3,4,5].

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
