# Peer review of "Clinical Implication of Drug Resistance for H. pylori Management"

_antibiotics, 2022, doi:10.3390/antibiotics11121684_

Round 1

Reviewer 1 Report

Generally speaking, this review article gave a comprehensive mechanism of H.pylori drug resistance and why we need antimicrobial susceptibility testing. I think all the listed topics and the structure are pretty clear. But I have a few revision suggestions.

1. It would be better if you could list all the resistance data and regional resistance data of all the drugs listed within a summary table and also insert the references in the table that would be a clear summary.

2. The context to use NGS to detect resistance is not quite clear, and needs more clarification about the development of NGS to detect resistance and are pros and cons of it. And needs more data or figures to illustrate.

3. Current major treatment for H.pylori is combination therapy. I noticed that you mention the combination therapies but if you could add another section discussing 2-3 specific combination treatment regimens for H.pylori and show the treatment efficacy figures, resistance data and etc. that would be better.

Author Response

  1. It would be better if you could list all the resistance data and regional resistance data of all the drugs listed within a summary table and also insert the references in the table that would be a clear summary.

Please see the attached documented we created that summarizes table regional resistance data of all drugs. 

  1. The context to use NGS to detect resistance is not quite clear, and needs more clarification about the development of NGS to detect resistance and are pros and cons of it. And needs more data or figures to.

The context to use NGS to detect resistance has been extensively discussed in this review including data for prior to primary treatment and in refractory cases. For further clarification please see the following references which has also been included in in this review: Graham DY, Moss SF. Antimicrobial Susceptibility Testing for Helicobacter pylori Is Now Widely Available: When, How, Why. Am J Gastroenterol. 2022;117(4):524-8 and Argueta EA, Alsamman MA, Moss SF, D'Agata EMC. Impact of Antimicrobial Resistance Rates on Eradication of Helicobacter pylori in a US Population. Gastroenterology. 2021;160(6):2181-3 e1.

The pros and cons of NGS have also been discussed in this review .  

  1. Current major treatment for H.pylori is combination therapy. I noticed that you mention the combination therapies but if you could add another section discussing 2-3 specific combination treatment regimens for H.pylori and show the treatment efficacy figures, resistance data and etc. that would be .

We have modified the Introduction section to add detailed information about current therapies (including a new Table, Table 1, please see submitted manuscript). We have not added efficacy data because they differ markedly in different populations around the world and are beyond the scope of this review. Resistance data are covered extensively later in our manuscript

Introduction now reads:

Helicobacter pylori (H. pylori) infection has a well-established role in peptic ulcer disease, gastric cancer, and mucosa-associated lymphoid tissue lymphoma. H. pylori eradication cures gastritis and can alter the progression of long-term complications, or recurrence of disease (1). Eradication therapies have evolved since H. pylori was first cultured 40 years ago. Early on, it was appreciated that single antimicrobials had very low rates of eradication but combination therapies were more successful. In the 1990s, proton pump inhibitors (PPIs) were also included in eradication regimens because acid suppression was found to increase eradication rates, in addition to benefiting ulcer healing.

Currently available therapies reflect the results of many clinical trials that have been performed without regard to susceptibility testing. The success rates of these therapies is highly variable around the world, reflecting differing population adherence, pharmacokinetics and antimicrobial susceptibility (Table 1). Rates of antimicrobial-resistance among H. pylori strains are increasing worldwide. In this paper, mechanisms and rates of resistance and implications in the treatment of H. pylori infections will be reviewed. The availability of rapid and accurate sequencing-based methods to determine antimicrobial resistance profiles among H. pylori strains and its benefits in choosing the optimal antimicrobials to include in treatment regimens is also discussed.

Reference:

Shiotani A, Roy P, Lu H, Graham DY.Helicobacter pylori diagnosis and therapy in the era of antimicrobial stewardship. Therap Adv Gastroenterol. 2021 Dec 21;14:17562848211064080. doi: 10.1177/17562848211064080.

Reviewer 2 Report

Dear Authors

Thank you very much for your manuscript submission. The topic of your review is very interesting; however a Major Revision is needed:

1. It is recommended to read and add the following paper to the References section of the manuscript to have a fruitful introduction section:

Advances in diagnosis and treatment of Helicobacter pylori infection. Acta Microbiol Immunol Hung. 2017 Sep 1;64(3):273-292. doi: 10.1556/030.64.2017.008. Epub 2017 Mar 6. PMID: 28263101.

2. As your manuscript is associated with drugs and drug resistance in H.pylori, it is recommended to add the characteristics of the mentioned drugs regarding their chemical structures, mechanism of action, pharmacology, biochemistry, etc.

In this regard it is recommended to read and add the following effective data bases to References section of the manuscript to have a fruitful manuscript (You can add an effective table to show these data and information):

https://go.drugbank.com/

https://www.ncbi.nlm.nih.gov/pccompound/

3. Your manuscript depicts drug resistance in H.pylori from different point of views. It is recommended to read and add the following papers to References section of the manuscript to have a fruitful review:

Helicobacter pylori Biofilm-Related Drug Resistance and New Developments in Its Anti-Biofilm Agents. Infect Drug Resist. 2022 Apr 5;15:1561-1571. doi: 10.2147/IDR.S357473. PMID: 35411160; PMCID: PMC8994595.

Helicobacter Pylori-Induced Gastric Infections: From Pathogenesis to Novel Therapeutic Approaches Using Silver Nanoparticles. Pharmaceutics. 2022 Jul 14;14(7):1463. doi: 10.3390/pharmaceutics14071463. PMID: 35890358; PMCID: PMC9318142.

Helicobacter pylori infection and antibiotic resistance - from biology to clinical implications. Nat Rev Gastroenterol Hepatol. 2021 Sep;18(9):613-629. doi: 10.1038/s41575-021-00449-x. Epub 2021 May 17. PMID: 34002081.

Biomaterials for Helicobacter pylori therapy: therapeutic potential and future perspectives. Gut Microbes. 2022 Jan-Dec;14(1):2120747. doi: 10.1080/19490976.2022.2120747. PMID: 36070564; PMCID: PMC9467593.

Multidrug-Resistant Helicobacter pylori Strains: A Five-Year Surveillance Study and Its Genome Characteristics. Antibiotics 2022, 11, 1391. https://doi.org/10.3390/antibiotics11101391

HopE and HopD Porin-Mediated Drug Influx Contributes to Intrinsic Antimicrobial Susceptibility and Inhibits Streptomycin Resistance Acquisition by Natural Transformation in Helicobacter pylori. Microbiol Spectr. 2022 Apr 27;10(2):e0198721. doi: 10.1128/spectrum.01987-21. Epub 2022 Mar 2. PMID: 35234510; PMCID: PMC9045298.

Helicobacter pylori Biofilm Confers Antibiotic Tolerance in Part via A Protein-Dependent Mechanism. Antibiotics (Basel). 2020 Jun 24;9(6):355. doi: 10.3390/antibiotics9060355. PMID: 32599828; PMCID: PMC7345196.

Exploration of the molecular mechanisms underlying the antibiotic resistance of Helicobacter pylori: A whole-genome sequencing-based study in Southern China. Helicobacter. 2022 Apr;27(2):e12879. doi: 10.1111/hel.12879. Epub 2022 Feb 6. PMID: 35124867.

Identification of multiple single-nucleotide variants for clinical evaluation of Helicobacter pylori drug resistance. Talanta. 2022 Jun 1;243:123367. doi: 10.1016/j.talanta.2022.123367. Epub 2022 Mar 3. PMID: 35298930.

Author Response

It is recommended to read and add the following paper to the References section of the manuscript to have a fruitful introduction section:

Advances in diagnosis and treatment of Helicobacter pylori infection. Acta Microbiol Immunol Hung. 2017 Sep 1;64(3):273-292. doi: 10.1556/030.64.2017.008. Epub 2017 Mar 6. PMID: 28263101.

Thank you very much for your feedback. This review includes several other similar references that highlight non-invasive and invasive diagnostic techniques as well as treatment treatment options.

As your manuscript is associated with drugs and drug resistance in H.pylori, it is recommended to add the characteristics of the mentioned drugs regarding their chemical structures, mechanism of action, pharmacology, biochemistry, .

In this regard it is recommended to read and add the following effective data bases to References section of the manuscript to have a fruitful manuscript (You can add an effective table to show these data and information):

https://go.drugbank.com/

https://www.ncbi.nlm.nih.gov/pccompound/

Thank you for the feedback. In our discussion of drug resistance, there is mention of biochemistry (e.g., point mutations affect protein structure), mechanism of action (e.g., discussion of how antibiotics are cytotoxic), and chemical structures (e.g., description of protein structure). The focus of the paper was on clinical management of H. pylori and to further delve into these topics would be out of the paper’s intended scope

Your manuscript depicts drug resistance in H.pylori from different point of views. It is recommended to read and add the following papers to References section of the manuscript to have a fruitful:

  • Helicobacter pylori Biofilm-Related Drug Resistance and New Developments in Its Anti-Biofilm Agents. Infect Drug Resist. 2022 Apr 5;15:1561-1571. doi: 10.2147/IDR.S357473. PMID: 35411160; PMCID: PMC8994595.
  • Helicobacter Pylori-Induced Gastric Infections: From Pathogenesis to Novel Therapeutic Approaches Using Silver Nanoparticles. Pharmaceutics. 2022 Jul 14;14(7):1463. doi: 10.3390/pharmaceutics14071463. PMID: 35890358; PMCID: PMC9318142.
  • Helicobacter pylori infection and antibiotic resistance - from biology to clinical implications. Nat Rev Gastroenterol Hepatol. 2021 Sep;18(9):613-629. doi: 10.1038/s41575-021-00449-x. Epub 2021 May 17. PMID: 34002081.
  • Biomaterials for Helicobacter pylori therapy: therapeutic potential and future perspectives. Gut Microbes. 2022 Jan-Dec;14(1):2120747. doi: 10.1080/19490976.2022.2120747. PMID: 36070564; PMCID: PMC9467593.
  • Multidrug-Resistant Helicobacter pylori Strains: A Five-Year Surveillance Study and Its Genome Characteristics. Antibiotics 2022, 11, 1391. https://doi.org/10.3390/antibiotics11101391
  • HopE and HopD Porin-Mediated Drug Influx Contributes to Intrinsic Antimicrobial Susceptibility and Inhibits Streptomycin Resistance Acquisition by Natural Transformation in Helicobacter pylori. Microbiol Spectr. 2022 Apr 27;10(2):e0198721. doi: 10.1128/spectrum.01987-21. Epub 2022 Mar 2. PMID: 35234510; PMCID: PMC9045298.
  • Helicobacter pylori Biofilm Confers Antibiotic Tolerance in Part via A Protein-Dependent Mechanism. Antibiotics (Basel). 2020 Jun 24;9(6):355. doi: 10.3390/antibiotics9060355. PMID: 32599828; PMCID: PMC7345196.
  • Exploration of the molecular mechanisms underlying the antibiotic resistance of Helicobacter pylori: A whole-genome sequencing-based study in Southern China. Helicobacter. 2022 Apr;27(2):e12879. doi: 10.1111/hel.12879. Epub 2022 Feb 6. PMID: 35124867.
  • Identification of multiple single-nucleotide variants for clinical evaluation of Helicobacter pylori drug resistance. Talanta. 2022 Jun 1;243:123367. doi: 10.1016/j.talanta.2022.123367. Epub 2022 Mar 3. PMID: 35298930.

Thank you for the feedback. The listed papers are either included in the paper, cover material already mentioned by other references within our paper, or are out of the scope of the paper. For example, Tshibangu-Kabamba et al. are cited in the paper but listed as a reference to review. Other listed references to review contain similar material to our cited papers. The first reference listed discusses biofilms which were discussed and referenced in other papers. The third reference listed is out of the scope of this paper because it mentions how drugs are manufactured. 

Round 2

Reviewer 2 Report

Dear Authors

Thank you very much for your explanation. Obviously, the suggested References provide complementary material to the manuscript; because, I believe that a Review should be effective to cover its aim at the highest level.

Author Response

Thank you very much for your understanding and for reviewing our work.